# Genetic Background and Clinical Features in Arrhythmogenic Left Ventricular Cardiomyopathy: A Systematic Review

**DOI:** 10.3390/jcm11154313

**Published:** 2022-07-25

**Authors:** Riccardo Bariani, Ilaria Rigato, Marco Cason, Maria Bueno Marinas, Rudy Celeghin, Kalliopi Pilichou, Barbara Bauce

**Affiliations:** 1Department of Cardiac, Thoracic, Vascular Sciences and Public Health, University of Padua, 35128 Padua, Italy; riccardo.bariani@unipd.it (R.B.); marco.cason@unipd.it (M.C.); maria.buenomarinas@unipd.it (M.B.M.); rudy.celeghin@unipd.it (R.C.); kalliopi.pilichou@unipd.it (K.P.); 2Azienda Ospedaliera/Universita’ di Padova, 35128 Padova, Italy; ilaria.rigato@aopd.veneto.it

**Keywords:** arrhythmogenic left ventricular cardiomyopathy, Desmoplakin, Filamin-C, Desmin, Phospholamban

## Abstract

In recent years a phenotypic variant of Arrhythmogenic cardiomyopathy has been described, characterized by predominant left ventricular (LV) involvement with no or minor right ventricular abnormalities, referred to as Arrhythmogenic left ventricular cardiomyopathy (ALVC). Different disease-genes have been identified in this form, such as Desmoplakin (DSP), Filamin C (FLNC), Phospholamban (PLN) and Desmin (DES). The main purpose of this critical systematic review was to assess the level of knowledge on genetic background and clinical features of ALVC. A search (updated to April 2022) was run in the PubMed, Scopus, and Web of Science electronic databases. The search terms used were “arrhythmogenic left ventricular cardiomyopathy” OR “arrhythmogenic cardiomyopathy” and “gene” OR “arrhythmogenic dysplasia” and “gene”. The most represented disease-gene turned out to be DSP, accounting for half of published cases, followed by FLNC. Overall, ECG abnormalities were reported in 58% of patients. Major ventricular arrhythmias were recorded in 26% of cases; an ICD was implanted in 29% of patients. A total of 6% of patients showed heart failure symptoms, and 15% had myocarditis-like episodes. DSP is confirmed to be the most represented disease-gene in ALVC patients. An analysis of reported clinical features of ALVC patients show an important degree of electrical instability, which frequently required an ICD implant. Moreover, myocarditis-like episodes are common.

## 1. Introduction

Arrhythmogenic cardiomyopathy (ACM) is an inherited and progressive cardiac disease first described in 1982 and characterized by fibro-fatty replacement of the right ventricular (RV) myocardium, which predisposes to the onset of ventricular arrhythmias that can even lead to sudden death, especially in young males [1,2]. Differently from original descriptions, which considered the left ventricular (LV) involvement usually mild and when relevant mainly due to a disease progression in association with advanced RV forms, in the last years it has become clear that LV involvement can occur in early stages of the disease, independently of or concurrently with RV involvement [3,4].

Recently, a phenotypic variant characterized by predominant LV involvement with no or minor RV abnormalities, also referred to as Arrhythmogenic left ventricular cardiomyopathy (ALVC), has been described [5]. In this form, a diagnosis can be challenging, and it is usually made in the presence of a subepicardial or ring-like late gadolinium enhancement (LGE) at cardiac magnetic resonance (CMR), prominent LV dilatation/dysfunction in the setting of relatively mild or absent right-sided disease, peculiar ECG features (inferolateral T-wave inversion, low QRS voltages-LQRSv) and ventricular arrhythmias of LV origin [5,6,7,8].

Furthermore, patients affected with ALVC can show an uncommon manifestation characterized by chest pain, troponin release, and 12-lead electrocardiogram abnormalities with normal coronary arteries. This clinical presentation has been defined as "hot phase" and enters into differential diagnosis with acute myocarditis [9,10,11].

The diagnosis of ALVC can be challenging as overlapping clinical features with other cardiac diseases can be present, primarily dilated cardiomyopathy (DCM) and myocarditis. Compared to DCM, patients with ALVC often show a high degree of electrical instability disproportionate to the impaired LV systolic function. Moreover, post-contrast CMR sequences in DCM mainly consist of patchy mid-myocardial LGE LV involvement, while in ALVC they are characterized by sub-epicardial LGE, often involving the inferior and lateral walls [12]. 

Different genes have been found to be related to ALVC, and the more frequent are Desmoplakin (DSP), Filamin C (FLNC), Phospholamban (PLN), and Desmin (DES); however, there is a prevalence of variants of genes already known to be related to inherited cardiac disease and of gene-elusive cases [4].

The aim of the present critical systematic review of the literature is to assess the level of knowledge on clinical and genetic features of ALVC.

## 2. Materials and Methods

### 2.1. Study Plan

This study was conducted according to PRISMA guidelines (http://www.prisma-statement.org/ (accessed on 23 April 2022)). A search was run in the PubMed [13], Scopus [14] and Web of Science [15] electronic databases for clinical studies that investigated genotyped ALVC patients. We collected published research using the following search items: “arrhythmogenic left ventricular cardiomyopathy” or “arrhythmogenic cardiomyopathy” and “gene” or “arrhythmogenic dysplasia” and “gene”. The ‘‘related articles’’ option on the PubMed homepage was also considered. No restriction about publication date was applied. Titles and abstracts of articles available in the English language were also evaluated. The full texts of the publications identified were screened for original data, and the references in the articles retrieved were checked manually for other relevant studies. The literature search has been updated to 23 April 2022.

### 2.2. Inclusion and Exclusion Criteria

Studies were included when the following general criteria were met: (1) articles were original reports; (2) reports were published in the English language; (3) studies included only patients who received the diagnosis of ALVC (based on 12 lead ECG, 2D-echocardiogram and CMR features or through autoptic examination) with genetic evaluation and reporting detailed clinical features of each patient. Editorials and reviews were excluded.

### 2.3. Data Extraction

Two of the authors (B.B. and R.B.) extracted the data from the selected articles. Disagreements were dealt with by discussion among the team members. In the case of studies reporting information on cardiomyopathies other than ALVC, only data about patients with this specific disease were considered for the purpose of this review. Details of the search process and study selection are shown in Figure 1. We also checked the references of the included studies and systematic reviews to identify additional studies that were not captured by our database searches. Information extracted from the studies included the title, name of the first author, year of publication, country of study population and qualitative description of the target population. Each included study was analyzed to extract all available data and ensure the eligibility of every single patient. For our review, we considered the following patient data: age at diagnosis, sex, the presence of a genetic variant with pathogenicity classification, ECG abnormalities (ST segment elevation, LQRSv, negative T waves in precordial or limb leads), major arrhythmic events (sudden death, ventricular fibrillation, sustained ventricular tachycardia, syncopal episodes), heart failure (HF), LV dilation/dysfunction at 2D-echocardiogram, subepicardial or ring-like LGE at CMR, ICD implantation, and myocarditis-like episodes with troponin release (“hot phases”). Studies describing a series of ALVC patients in which it was not possible to assess individual clinical, instrumental, and genetic features were excluded from the analysis.

## 3. Results

### 3.1. Retrieving Studies

A total of 3991 titles were retrieved (1729 from PubMed, 1010 from Scopus, and 1252 from Web of Science). After removing duplicates, a total of 2395 titles were screened, allowing us to identify 87 studies potentially relevant to the topic. The full-text screening of these articles led to the exclusion of 56 studies due to their compliance with the inclusion/ exclusion criteria. The remaining 31 articles were considered eligible for this review [10,16,17,18,19,20,21,22,23,24,25,26,27,28,29,30,31,32,33,34,35,36,37,38,39,40,41,42,43,44,45] A PRISMA flow diagram depicts the flow of information through the different literature review phases (Figure 1). Our inclusion criteria, with regard to the presence of detailed patients’ clinical features, led to the exclusion of several publications reporting a significant number of subjects. In addition, even in selected studies, the ECG description was sometimes poor and the presence of LQRSv was not emphasized, especially in older studies when this peculiar ECG pattern could have not been reported.

### 3.2. Disease Genes in ALVC

The majority of studies describing patients affected with ALVC reported the presence of DSP mutation (n = 20, 65% of revised studies), followed by FLNC (n = 4), PLN (n = 3) and LMNA (n = 1), (DES (n = 2). The presence of the DSG2 and PKP2 genetic variants were reported in two case reports (one for each disease gene). Overall, ECG abnormalities were detected in 58% of patients, with LQRSv and negative T waves in lateral and inferior leads being the most common abnormalities (in 26% and 36% of published cases, respectively). An LV dilatation/dysfunction was reported in 61% and the presence of LV epicardial LGE was reported in 79% of cases. Major ventricular arrhythmias were detected in 26% of patients; an ICD was finally implanted in 29% of cases. A total of 6% of reported cases showed heart failure signs and symptoms and 15% experienced a myocarditis-like episode. A summary of the 31 selected studies is reported in Table 1.

## 4. Discussion

### 4.1. ALVC Linked to DSP Genetic Variants

Plakin proteins form cell-cell and cell-matrix junctions and link to organelles by engaging intermediate filaments, actin microfilaments and microtubules. Plakin proteins are widely distributed in tissues including the epithelia, cardiac muscle and skeletal muscle and mediate specialized functions [46]. Desmoplakin (DSP) represents an important member of this family of proteins, and it is essential for cell-cell adhesion in desmosomes due to their essential role in maintaining tissue integrity and resilience; compromised plakin function can lead to genetic and autoimmune diseases. A homozygous deletion in DSP (DSP6901delG), which results in a premature stop codon and a truncated protein product lacking the C-domain of the tail region was for the first time linked to cardiomyopathy in patients with Carvajal disease, a cardio-cutaneous syndrome characterized by dilated cardiomyopathy (DCM), palmoplantar keratoderma and woolly hair [47]. A few years later, an association between DSP genetic variants and ACM was reported [48]. Interestingly, the pathology investigation of a heart specimen with Carvajal syndrome demonstrated biventricular dilatation and diffuse scarring of the free walls of both the right ventricle (RV) and the left ventricle (LV), with areas of extensive myocardial loss and replacement fibrosis in the outer third of the LV [49]. In the following years, several studies demonstrated a correlation between ALVC and variants in the DSP gene. Smith et al. (2020) collected a series of patients carrying DSP truncating mutations, proposing the term “desmoplakin cardiomyopathy” to describe a clinical phenotype characterized by a large amount of LV fibrosis, episodes of myocardial necrosis and a significant degree of electrical instability, and who entered into differential diagnosis with both ACM and DCM [11]. Recently, Bariani et al. (2021) collected a series of 21 ACM patients, showing an uncommon clinical presentation of the disease characterized by chest pain, ECG abnormalities and troponin release that has been defined as the ‘Hot phase’ [10]. Of these, 19% were DSP carriers with the ALVC phenotype. Confirming previous observations on DSP prevalence in the ALVC phenotype, our analysis demonstrated that published studies describing the genetic background and clinical features of ALVC patients in detail reported the presence of DSP genetic variants in 57% of cases (Table 1). In our review, we found that in 80 DSP variant carriers (39% probands, 45% males, mean age at diagnosis 38 ± 20 yrs), ECG abnormalities were present in 51% of cases with typical ALVC findings, and LQRSv and negative T wave in lateral and inferior leads in 14% and 29% of cases, respectively. Major ventricular arrhythmias (MAV) occurred in 22% and HF occurred in 6% of patients. Chest pain episodes with troponin release were observed in 15% of cases.

### 4.2. ALVC Linked to FLNC Genetic Variants

Filamin C (FLNC) is an important structural crosslinker of actin rods at the sarcomeric z-disc of both cardiac and skeletal muscle. Moreover, as filamin A, FLNC can serve as a nodal point for sarcomeric mechano-transduction in different muscle cells [50]. Filamin C was first reported to be associated with various forms of skeletal myopathy [51]. Truncating FLNC mutations have been identified in DCM patients. Consistently with other genetic variants found in cardiomyopathy, FLNC variants found in human DCM are not accompanied by concomitant myofibrillar myopathy. In 2016, Ortiz-Genga identified 23 new truncating variants of FLNC in a DCM cohort. Moreover, the FLNC-DCM phenotype was found to show a marked LV-dilation and systolic dysfunction, a high degree of myocardial fibrosis and ECG abnormalities (T-Wave changes and LQRSv) [23]. Interestingly, Begay et al. (2016) described a phenotypic RV involvement in a series of FLNC truncated mutation carriers, thus indicating a potential phenotypic overlap between DCM and ACM in some FLNC mutation carriers [52]. It is noteworthy that truncating FLNC variants have been rarely reported in patients affected with the “classical” form of ACM [53]. Celeghin et al. (2021) reported a series of ACM probands that tested negative for mutations in ACM-related genes which underwent FLNC genetic screening; novel FLNC variants were detected in 4% of patients. Clinical evaluation found that the most common ACM disease phenotype was ALVC and that patients were characterized by a late disease onset (after 40 years). In addition, FLNC-associated cardiomyopathy was characterized by ECG abnormalities such as LQRSv and inferolateral negative T waves, frequent and complex VAs, and extensive nonischemic LV LGE/myocardial fibrosis on CMR or postmortem analysis [41]. Gigli et al. (2021) recently analyzed a large series of FLNC variant carriers [54] and found an ALVC phenotype in 21% of cases, a DCM phenotype in 42% and an ARVC phenotype in 3%. In our analysis on published studies describing the genetic background and clinical features of ALVC patients in detail, a FLNC was the second ALVC related gene in term of patient numbers (35 pts, 46% probands, 71% males, mean age at diagnosis 46 ± 18 yrs), accounting for 25% of reported cases. (Table 1). In our review, we found that in FLNC variant carriers, ECG abnormalities were common (54% of cases) with the presence of LQRSv and negative T waves in lateral and inferior leads, in 23% and 40% of cases, respectively. MVA occurred in 31% and HF occurred in 6% of patients. Overall, 11% of patients received an ICD. Chest pain episodes with troponin release were observed in 9% of cases.

### 4.3. ALVC Linked to DES Genetic Variants

DES plays key structural and signaling roles in myocytes and is critical for cytoskeletal organization and maintaining cardiomyocyte structure [55]. Originally, a Desmin-related myopathy (DRM) was described and characterized by myopathy, often associated with a wide spectrum cardiac involvement, primarily DCM [56,57,58]. Indeed, a meta-analysis of 159 patients with 40 different DES mutations reported in the literature [59] indicated that up to 50% of carriers had cardiomyopathy, mostly DCM (17%), restrictive cardiomyopathy (12%), hypertrophic cardiomyopathy (6%), and rarely ACM (1%). A first association between mutations in DES and ACM was suggested by Van Tintelen et al. in 2009 [60]. The authors investigated the clinical-instrumental findings of five probands and 17 family members carrying the 38C>T mutation in DES. They concluded that the phenotype was highly variable with predominantly cardiological clinical pictures of right-sided myocardial involvement (also in keeping with the diagnosis of ARVC). One year later, Klauke et al. (2010) confirmed the association with the disease through both clinical and functional characterization [61]. Subsequently, Otten et al. (2010) confirmed this association and stated that one of the possible cardiac phenotypes of DES mutations was a biventricular cardiomyopathy with right ventricular features in keeping with ACM [55]. Finally, Bermúdez-Jiménez et al. (2018) described a large family in which approximately 30 members affected with an ACM phenotype harbored a missense pathogenic variant of the DES (p.Glu401Asp). In detail, all subjects carrying the mutations presented phenotypic features in keeping with ACM, with almost exclusive left ventricular involvement at CMR, a high incidence of ventricular arrhythmias and sudden cardiac death in the absence of conduction alterations and peripheral myopathy. Moreover, no episodes of “hot phases” were reported [24]. In our analysis we found two studies describing DES variant carriers who met the diagnosis of ALVC; overall clinical and instrumental features of 22 patients were reported in detail (16% of our cohort, 9% probands, 41% males, mean age at diagnosis 48 ± 17 yrs). An analysis of ECG features revealed the presence of abnormalities in 86%, with LQRSv and negative T wave in lateral and inferior leads recorded in 41% and 77% of cases, respectively. Major ventricular arrhythmias occurred in 59% of patients and HF in 18%. An ICD was implanted in 45% of subjects. Chest pain episodes with troponin release were observed in 18% of reported cases.

### 4.4. ALVC Linked to PLN Genetic Variants

PLN has a key role in the function of the sarcoplasmic reticulum, as it acts as a regulator enabling calcium transport through the Ca^2+^-ATPase pump (SERCA2a). The phosphorylation of PLN plays a key role in calcium transport through SERCA2A. Indeed, in its de-phosphorylated form it acts as an inhibitor of SERCA2A, while phosphorylation eliminates this inhibition and allows an increase in calcium accumulation in the sarcoplasmic reticulum [62]. Among the pathogenic variants identified, R14del is the most common within the cohorts of patients affected by DCM and ACM, particularly in The Netherlands, where it reaches a prevalence of 10–15% in ACM patients [63]. Clinical-instrumental findings of these patients are low-voltage ECG, a high frequency of malignant ventricular arrhythmias, and end-stage heart failure [64]. Van Rijsingen et al. (2014), through the study of a cohort of 403 patients carrying the R14del mutation, observed that during a follow-up period of approximately four years, 19% of patients presented a malignant ventricular arrhythmia, while 11% had end-stage heart failure. In addition, the authors highlighted the role of left ventricular ejection fraction < 45% and the presence of non-sustained ventricular tachycardia on Holter ECG as independent predictors of malignant arrhythmias [64]. Recently, Verstraelen et al. (2021) proposed a risk score incorporating new clinical parameters such as premature ventricular contraction count/24 h, amount of negative T waves, and the presence of LQRS v [65]. In our analysis, we identified two case series and one case report describing PLN variant carriers showing the ALVC phenotype in detail (10 pts, 7% of our cohort) (Table 1).

### 4.5. ALVC Linked to LMNA Variants

LMNA encodes a nucleo-skeletal intermediate filament with complex cellular functions, including maintaining nuclear structural integrity, regulating gene expression, mechanosensing, and mechano transduction through the lamina-associated proteins [66]. Pathogenic variants of LMNA are associated with a wide spectrum of diseases, including muscular dystrophies (e.g., Emery-Dreifuss), Hutchinson-Gilford Progeria Syndrome, and cardiac manifestation [67]. Among the latter, LMNA genetic variants have been firstly reported in patients with DCM frequently associated with conduction disturbances and a high degree of arrhythmic instability which does not correlate with LV systolic function. This evidence led to the indication for ICD implantation in patients carrying a pathogenic variant of the LMNA gene showing a value of LV-EF below 45% in the presence of risk factors [68]. On the other hand, its putative role as an ACM-causing gene is still limited and debated [69]. Quarta et al. (2012) described four patients carrying an LMNA genetic variant who showed a cardiac phenotype in keeping with classical ARVC [70]. The clinical instrumental features of these patients were a family history of cardiomyopathy and/or sudden death, T-wave inversion in precordial leads, and atrioventricular (AV) and/or intraventricular conduction delays at ECG, while an RV dilation/dysfunction was present in three out of four patients. Other publications in the following years reported similar findings [71,72]. Regarding ALVC, we identified only one paper reporting detailed clinical and instrumental findings of three young LMNA carriers fulfilling ALVC criteria; notably, two of them underwent cardiac transplant due to refractory HF [23].

### 4.6. ALVC Linked to Other Disease Genes

ALVC seems to be rarely reported in patients carrying genetic variants of the desmosomal gene with the exception of DSP. In our review we found only one PKP2 carrier and three DSG2 carriers showing the ALVC phenotype. Sen-Chowdhry et al. (2008) described the clinical and instrumental features of an ALVC cohort of 42 patients belonging to 24 families [5]. In eight families, a desmosomal genetic variant was found with a predominance of DSP, but also the presence of PKP2 c.419C4T) and DSG2 genetic variants in two different patients has been reported [5]. Regarding the PKP2 c.419C4T variant, several studies have analyzed and questioned its pathogenicity, although in no case an association with ALVC been reported [73,74]. Similarly, Smith et al., (2020) among 79 PKP2 gene carriers, did not find any ALVC form [11]. In addition, Casella et al. (2020) evaluated a series of ALVC patients and among 12 with a positive genetic result found three DSG2 and 9 DSP genetic variant carriers [75]. Recently, Graziosi et al. (2022) described a large series of ALVC patients and reported the presence of genetic variants of SCN5A and TMEM43 genes [76].

## 5. Conclusions

DSP is confirmed to be the most represented disease-gene in ALVC patients. The analysis of reported clinical features of ALVC patients show an important degree of electrical instability which frequently required an ICD implant. Moreover, myocarditis-like episodes are common.

## 6. Limitations

The study has some limitations. As we decided to consider only studies in which genetic, clinical and instrumental features of each patient were reported, this may have resulted in the loss of some data. Furthermore, to date, the role of individual genes in the pathogenesis of left-dominant arrhythmogenic cardiomyopathy remains largely unknown. Future studies are necessary to better understand the possible interactions between genes and the intracellular molecular pathways also in order to explore new therapeutic options.

## Figures and Tables

**Figure 1 jcm-11-04313-f001:**
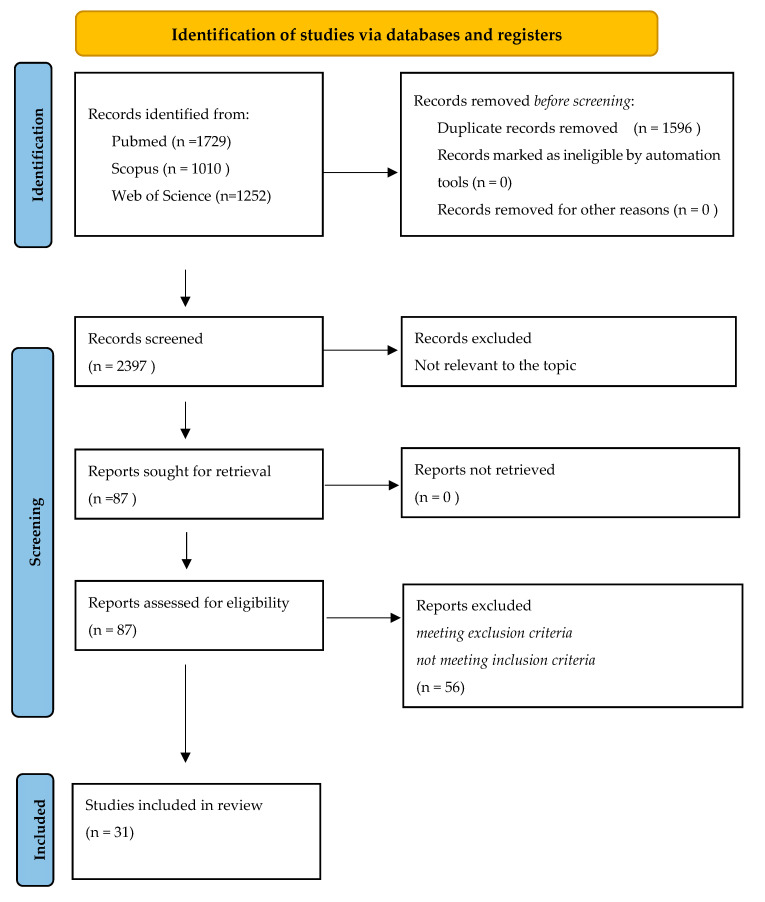
PRISMA flow diagram summarizing the literature review and inclusion/exclusion process.

**Table 1 jcm-11-04313-t001:** Studies included in the systematic review.

Author/Year	Type of Publ	N. of Pts	Age at Diagnosis (yrs)	M (%)	Disease Gene	ECG abn (n)	LV Echo Dilation /Dysfuntion (n)	LV Epicardial LGE (n)	Hot Phase (n)	MVA (n)	ICD (n)
Norman et al., 2005 [16]	Original research	12	39 ± 14	4	DSP	7/12	10/12	8/12	0	4/12	4/12
Posch et al., 2009 [17]	Original research	5	32 ± 8	3	PLN	5/5	5/5	2/2	0	3/5	1/5
Navarro-Manchón et al., 2011 [18]	Brief Report	4	50 ± 19	2	DSP	¾	¼	2/4	0	¼	0
Pilichou et al., 2014 [19]	Brief Report	3	31 ± 20	2	DSP	3/3	0	3/3	0	1/3	0
López-Ayala et al., 2014 [20]	Original research	6	38 ± 18	3	DSP	NA	NA	6/6	0	2	6
Saguuner et al., 2015 [21]	Case Report	1	36	1	PKP2	1/1	0	1/1	0	1/1	1
López-Ayala et al., 2015 [22]	Reseach letter	4	45 ± 23	1	PLN	2/4	2/4	2/3	0	¼	1
Ortiz-Genga et al., 2015 [23]	Original research	15	40 ± 17	11	FLNC	6/12	10/13	13/14	3/15	6/15	1/15
Bermudez-Jimenez et al., 2018 [24]	Original research	13	43 ± 18	4	DES	10/13	11/13	13/13	4/13	10/13	3/13
DeWitt et al., 2019 [25]	Original research	7	13 ± 2	4	DSP, DSP + DSG2, LMNA	NA	7/7	5/7	2/7	4/7	0
Li et al., 2019 [26]	Case Report	1	36	1	PLN	1	1	1	0	0	0
Piriou et al., 2020 [27]	Original research	16	40 ± 24	6	DSP, DSP + MYBPC3	6/15	5/15	14/16	2	1	NA
Tsuruta et al., 2020 [28]	Case Report	1	74	1	DSP	1	1	1	0	0	1
Hall et al., 2020 [29]	Original research	10	58 ± 18	6	FLNC	7/10	6/10	9/10	0	2/10	0
Verma et al., 2020 [30]	Case Report	1	21	0	DSP	NA	1	0	1	0	0
Kissopoulou et al., 2020 [31]	Brief Reports	3	25 ± 3	2	DSP	2	1	3	2/3	0	0
Poller et al., 2020 [32]	Case Report	1	15	1	DSP + Dystrophin	1	1	1	1	0	0
Graziosi et al., 2020 [33]	Case Report	2	19, 46	2	DSP	0	0	1	0	1	1
Maghin et al., 2019 [34]	Case Report	1	16	0	DSP	NA	NA	0	0	1	0
Heliö et al., 2020 [35]	Original research	12	45 ± 17	7	DSP	10	11	4/4	1	5	6
Protonotarios et al., 2020 [36]	Original research	9	52 ± 13	5	DES	9/9	8/9	7/8	0	2/9	7/9
Leite et al., 2021 (37)	Case Report	2	16, 61	2	DSP	½	½	2/2	0	2/2	0
Lao et al., 2021 [38]	Case Report	1	28	0	DSG2	1	0	1	0	1	1
Kandhari et al., 2021 [39]	Case Report	1	50	1	FLNC	1	1	1	0	1	1
Rubino et al., 2021 [40]	Case Report	3	36 ± 17	1	DSP	3/3	3/3	3/3	1/3	0	1/3
Celeghin et al., 2021 [41]	Original research	9	40 ± 17	7	FLNC	5/8	4/8	6/6	0	3/9	2/9
Bariani et al., 2021 [10]	Original research	4	23 ± 21	2	DSP	2/4	¼	4/4	3/4	0	0
Rawal et al., 2021 [42]	Case Report	3	50 ± 27	1	DSP, DSP + JUP	NA	NA	3/3	0	1	3
Efthimiadis et al., 2021 [43]	Case Report	3	47 ± 26	0	DSP	2/3	2/3	2/3	1	1	1
Westphal et al., 2022 [44]	Case Report	2	24, 25	1	DSP	NA	1/1	2/2	1/2	1/1	1
Santos Ferreira et al., 2022 [45]	Case Report	1	45	1	DSP	1	1	1	0	1	1

publ: publication; ALVC: Arrhythmogenic left ventricular cardiomyopathy; M: males; SD: sudden death; ECG abn: ECG abnormalities; LV: left ventricular; LGE: late gadolinium enhancement; ICD: implantable defibrillator; NA: not available; DSP: Desmoplakin; PLN: Phospholamban; PKP2: Plakofillin-2; DES: Desmin; DSG2: Desmoiglein-2; FLNC: Filamin-C, LMNA: laminin-A, JUP: junction Plakoglobin, MYBPC3: cardiac myosin binding protein C, MVA: Major Ventricular Arrhythmias.

## Data Availability

The data presented in this study are available on request from the corresponding author.

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
