# Peer review of "Genetic Background and Clinical Features in Arrhythmogenic Left Ventricular Cardiomyopathy: A Systematic Review"

_jcm, 2022, doi:10.3390/jcm11154313_

Round 1

Reviewer 1 Report

The data is a bit misleading and perhaps missing. Studies with an ACM diagnosis (which could be ARVC OR ALVC) are not included. AND there is a possibility of misdiagnosis of ARVC with LV involvement (also leading to missing data). It is important to at least acknowledge these facts.

Table 1 may be easier to read is turned horizontally on the paper. Data from table 1 is hard to understand because it is so condensed with the wording. Also missing units for some numbers.

It would be helpful to add a discussion hypothesizing/speculating the roll that the 5 gene products play in the phenotypes of ALVC. How are the genes related? Do they share commonalities? Can any molecular pathway be identified by their link to the disease? This type of discussion would greatly enhance the discussion of the paper. In addition, it might be important to add a future directions section, highlighting areas that are lacking in the field and where the field needs to go.

In the discussion section 1, plakin proteins are introduced then DSP is discussed. But it is not stated that DSP is a plakin protein.

What is LDAC? Page 6 line 187

It is unclear what myocardial enzyme release is? What enzyme is released?

In the discussions about DSP, desmin, and lamin A nothing is noted about the gene…how many exons? Where is it located? Etc However, this info is included for Filamin C. Either the info should be included for all or left out for all. 

SERCA2A PLN interaction does not determine the rate on contraction as stated on page 8 line 272

Author Response

The data is a bit misleading and perhaps missing. Studies with an ACM diagnosis (which could be ARVC OR ALVC) are not included. AND there is a possibility of misdiagnosis of ARVC with LV involvement (also leading to missing data). It is important to at least acknowledge these facts.

Thank you for your comment. In the article selection phase, we decided to consider only manuscripts reporting studies on patients who had been diagnosed with ALVC (based on 12-lead ECG, 2D echocardiogram, and CMR features or by autopsy examination) in whom genetic results and detailed clinical findings of each patient were reported. Thus, we did not consider studies reporting patients with right ventricular involvement that wan not in keeping with the definition of left-dominant arrhythmogenic cardiomyopathy.

Certainly, the strict selection process led to the exclusion of some studies that did not reported in detail patients clinical and instrumental findings.

Table 1 may be easier to read is turned horizontally on the paper. Data from table 1 is hard to understand because it is so condensed with the wording. Also missing units for some numbers.

Thank you for the suggestion. Table was rotated and expanded. In addition, all units of measurement were reported.

It would be helpful to add a discussion hypothesizing/speculating the roll that the 5 gene products play in the phenotypes of ALVC. How are the genes related? Do they share commonalities? Can any molecular pathway be identified by their link to the disease? This type of discussion would greatly enhance the discussion of the paper. In addition, it might be important to add a future directions section, highlighting areas that are lacking in the field and where the field needs to go.

Thank you for the suggestion. To date, evidence on the pathogenesis of Arrhythmogenic cardiomyopathy is limited. Moreover, the role of specific genes is controversial and debated, as the same gene can lead to different phenotypes. Certainly, future studies are needed to clarify this important aspect of the disease.

A limitation section reporting these considerations was added.

In the discussion section 1, plakin proteins are introduced then DSP is discussed. But it is not stated that DSP is a plakin protein.

Thank you for your comment. The sentence has been revised to clarify that Desmoplakin is part of the plakin protein family. Changes have been made in the text on page 8, lines 178-179.

What is LDAC? Page 6 line 187

Thank you for your comment. It was a typo. The term LDAC was replaced with ALVC at page 8 line 191.

It is unclear what myocardial enzyme release is? What enzyme is released?

Thank you for your comment. We have replaced the term myocardial enzyme release with troponin release. Changes have been made to the text on page 2, line 48; page 8, line 198; page 9, line 265.

In the discussions about DSP, desmin, and lamin A nothing is noted about the gene…how many exons? Where is it located? Etc However, this info is included for Filamin C. Either the info should be included for all or left out for all. 

Thank you. Following reviewer’s suggestion, description of the FLNC gene was modified.

SERCA2A PLN interaction does not determine the rate on contraction as stated on page 8 line 272

Thank you. The sentence was modified in the text on page 10, lines 277-279

Moreover, the English language of the revised version of the manuscript has been supervised by a native speaker professional translator (see Acknowledgments)

Reviewer 2 Report

The authors present a clear review of genetic background and clinical features in arrhythmogenic left ventricular cardiomyopathy

The review is well-structured and I believe the paper to be of interest especially to the non-specialist in the field of cardiomyopathy and is of educational value.

However, probably due to  selected  search items a few published papers were not included into analysis, for example (1) Graziosi M, Ditaranto R, Rapezzi C, et al. Clinical presentations leading to arrhythmogenic left ventricular cardiomyopathy. Open Heart 2022;9:e001914. (first published April 20, 2022) or Ma C, Fan J, Zhou B. et al. Myocardial strain measured via two-dimensional speckle-tracking echocardiography in a family diagnosed with arrhythmogenic left ventricular cardiomyopathy. Cardiovasc Ultrasound 19, 40 (2021) (published 20 December 2021).

Thus,  mutations in a few less common genes (SCN5A, TMEM43) were not listed as causing ALVC.

It should be mentioned in limitations section.

In the abstract section please put a period at the end of the sentence

(page 1,  line 16).

Author Response

The authors present a clear review of genetic background and clinical features in arrhythmogenic left ventricular cardiomyopathy. The review is well-structured and I believe the paper to be of interest especially to the non-specialist in the field of cardiomyopathy and is of educational value.

However, probably due to  selected  search items a few published papers were not included into analysis, for example (1) Graziosi M, Ditaranto R, Rapezzi C, et al. Clinical presentations leading to arrhythmogenic left ventricular cardiomyopathy. Open Heart 2022;9:e001914. (first published April 20, 2022) or Ma C, Fan J, Zhou B. et al. Myocardial strain measured via two-dimensional speckle-tracking echocardiography in a family diagnosed with arrhythmogenic left ventricular cardiomyopathy. Cardiovasc Ultrasound 19, 40 (2021) (published 20 December 2021). Thus,  mutations in a few less common genes (SCN5A, TMEM43) were not listed as causing ALVC. It should be mentioned in limitations section.

Answer: Thank you for your comment. The selection process with inclusion of papers in which detailed genetic, clinical and instrumental findings of each patient were reported led to the exclusion of very interesting studies. However, considering that the paper of Graziosi et al described a large cohort of ALVC patients and reported the presence of mutations of less common genes, the study was reported among references and commented in the discussion chapter (page 11 lined 329-330, ref 76).

In the abstract section please put a period at the end of the sentence (page 1,  line 16).

Answer: thank you. The typo was correct in the text: page 1, line 16.